# Assessing the Validity and Reliability of A Low-Cost Microcontroller-Based Load Cell Amplifier for Measuring Lower Limb and Upper Limb Muscular Force

**DOI:** 10.3390/s20174999

**Published:** 2020-09-03

**Authors:** Julie Gaudet, Grant Handrigan

**Affiliations:** Faculté des Sciences de la Santé et des Services Communautaires, École de Kinésiologie et de Loisir, Université de Moncton, Moncton, NB E1A 3E9, Canada; ejg9116@umoncton.ca

**Keywords:** muscular strength, microcontroller, criterion validity, inter-day reliability, force, isometric

## Abstract

Lower and upper limb maximum muscular force development is an important indicator of physical capacity. Manual muscle testing, load cell coupled with a signal conditioner, and handheld dynamometry are three widely used techniques for measuring isometric muscle strength. Recently, there is a proliferation of low-cost tools that have potential to be used to measure muscle strength. This study examined both the criterion validity, inter-day reliability and intra-day reliability of a microcontroller-based load cell amplifier for quantifying muscle strength. To do so, a low-cost microcontroller-based load cell amplifier for measuring lower and upper limb maximal voluntary isometric muscular force was compared to a commercial grade signal conditioner and to a handheld dynamometer. The results showed that the microcontroller-based load cell amplifier correlated nearly perfectly (Pearson's R-values between 0.947 to 0.992) with the commercial signal conditioner and the handheld dynamometer, and showed good to excellent association when calculating ICC scores, with values of 0.9582 [95% C.I.: 0.9297–0.9752] for inter-day reliability and of 0.9269 [95% C.I.: 0.8909–0.9533] for session one, intra-day reliability. Such results may have implications for how the evaluation of muscle strength measurement is conducted in the future, particularly for offering a commercial-like grade quality, low cost, portable and flexible option.

## 1. Introduction

Muscle strength has been defined as the maximum force (in N) or torque (in Nm) developed during maximal voluntary contraction under a given set of conditions [1]. Therefore, the term “muscular strength” typically refers to a measure describing an individual’s ability to exert maximal muscular force, either statically or dynamically [2]. In all circumstances, an activated muscle develops force. Depending on the interaction between the force developed by the muscle and the load on the muscle, the muscle will either shorten, remain at a fixed length, or be lengthened [3]. Tension development, without a change in muscle length is an isometric muscle action [4]; the muscle acts against an immovable resistance at a specific joint angle [2]. Tension development with changes in the length of the muscle, allowing for a complete range of motion, is an isotonic muscle action [2]. An isotonic-concentric action involves the development of tension during shortening of the muscle, whereas an isotonic-eccentric action occurs when there is development of tension during lengthening of the muscle [4].

Muscle strength tests are widely used in sports, physical education, ergonomic practice and clinical practice [1]. The ability to assess muscle function is a key aspect of the physical capacity evaluation in guiding interventions targeted at improving muscular strength [5]. Manual muscle testing, load cell in combination with a signal conditioner and a handheld dynamometry are three techniques that are widely used for measuring isometric muscle strength. Recently, there is a proliferation of low-cost tools that have potential to be used to measure muscle strength in a variety of applications, including portable and wearable options. 

Given that muscle strength is defined as maximum voluntary development by a particular muscle group [6], it is accepted that maximum muscle contraction should be the subject of analysis for any given measurement tool, and maximum voluntary isometric contraction (MVIC) appears to be the most common method of standardization [7]. In isometric strength testing, the muscle acts against an immovable resistance at a specific joint angle and does not change length [2]. According to Croisier and Crielaard (1999), maximum isometric contraction appears to be reproducible, however, various factors must be considered in conducting isometric testing. These factors, include the joint angle at which the test is performed, the rest interval between two consecutive repetitions, the number of repetitions performed, and the body position. A lack of standardization across studies for these factors limits the ability to compare results [8]. This highlights the importance of following and reporting standardized assessment procedures, in order to obtain the most valid and reliable results with any given measurement instrument for MVIC. 

There are several methods to measure an individual's muscle strength and the choice of technique affects the muscle strength values obtained. The manual muscle test (MMT) is the most commonly used method for documenting muscle strength disorders in clinical settings [9]. This method involves observation, palpation and forced application by an examiner to determine the strength of a muscle action [10]. Of concern, MMT is affected by the judgment and strength of the assessor and field assessments cannot be completed by extremely weak patients [10], thus, making this a muscle strength measurement technique of limited value. Fortunately, there are tools that provide more objective and accurate results. Commonly referred to as the “golden standard”, are the isokinetic dynamometers (ID) such as the Cybex II isokinetic dynamometer [11,12]. However, their clinical use is limited, due to their high cost, non-portability, and their complicated setup and operation [12]. Similarly, the load cell, coupled with a signal conditioner, is a laboratory setup commonly used to accurately measure isometric muscle strength. To record these values, an analog to digital setup and a computer are necessary, thus, making this method expensive and not portable. Fortunately, current technological advances open the door to many portable, fast and easy to perform methods for quantifying muscle strength. A commonly used compromise between MMT and ID devices is a portable device known as the handheld dynamometer [12], a clinical instrument that quantifies isometric strength. However, it is known that measurements obtained with the handheld dynamometer are influenced by the practitioner's strength [13]. Therefore, researchers recommend the use of a belt to stabilize the device [14] and offering improved measurement. The second portable device is a microcontroller-based signal amplifier, such as the HX711 (Sparkfun, Niwot, CO, USA), and when paired with a load cell and a microcomputer, it potentially offers a low-cost customizable measure of force. This system is small, lightweight, and consumes little power. It can be programmed for different weight ranges with some minor modifications in software and hardware. That said, calibration is of critical importance for the proper operation of the instrument and thus for obtaining valid and reliable measurements [15]. Nevertheless, microcontrollers are becoming more and more available at low cost and their use is therefore becoming more accessible. However, before systems using these microcontrollers can be implemented, their validity and reliability must be examined. Precisely, the criterion validity, the intra-day reliability and the inter-day reliability of a low-cost microcontroller-based load cell amplifier for measuring maximum isometric muscle strength development, in the context of force assessments has not been studied widely, and therefore needs to be explored [16]. 

The primary aim of this study was to assess the criterion validity, intra-day reliability and inter-day reliability of a low-cost microcontroller-based load cell amplifier for measuring upper and lower limb maximal isometric muscular force by comparing it to a standard laboratory setup (e.g., commercial grade signal conditioner and load cell) and to a portable handheld dynamometer. Voluntary maximum isometric muscle contraction was chosen because it allowed for simultaneous assessment of strength from the low-cost microcontroller-based load cell amplifier, the standard laboratory setup, and the clinical device, and could be easily replicated over two sessions. We hypothesized a strong agreement in the validity of the two pairs of instruments, namely between the microcontroller-based load cell amplifier and the handheld dynamometer, as well as the microcontroller-based load cell amplifier with the standard laboratory setup. We also hypothesize strong inter-day reliability and intra-day reliability for all three instruments. 

## 2. Materials and Methods

The study was approved by the Ethics Committee of the Université de Moncton (project ID 1819-087) in accordance with the standards established by the Declaration of Helsinki of 1975, revised in 2013. Exclusion criteria included being under 18 years of age and an injury or discomfort preventing the participants from performing repeated maximal voluntary contractions of their dominant arm and leg. Participants went to the CEPS at the Université de Moncton for two testing sessions on two separate days. Participants received and signed a copy of the consent form prior to the start of the testing sessions. Each testing session lasted approximately 45 minutes and consisted of four trials of maximum isometric strength development of the lower limb (i.e., leg extension), and four trials of maximum isometric strength development of the upper limb (i.e., biceps flexion). Each condition was presented in randomized order. Participants were asked to identify their dominant leg and arm, which was used as the limbs to be measured in both sessions.

Once informed consent was obtained, each participant was simultaneously equipped with a handheld dynamometer (MicroFET 2, Lafayette, IN, USA) as well as a load cell (CZL301C, Phidgets Inc., Calgary, AB, Canada) connected to a commercial signal conditioner (2100, Vishay, Shelton, CT, USA) and a low-cost microcontroller-based amplifier. The signal conditioner was connected to an analog to digital card (USB-1616FS, MCC, Norton, MA, USA) and a computer. These three tools represent three systems that measure muscle strength. Price comparison and characteristics of the three tools are presented in Table 1. In this experiment, the load cell coupled with the commercial signal conditioner is referred to as the laboratory setup and holds the standard measurements of all three devices. The same muscle contractions were measured for each of the instruments in order to minimize the measurement error that occurs between different trials during a maximum voluntary contraction. Each participant was therefore equipped with these three tools at the forearm level for the upper limb measurement and at the tibial level for the lower limb measurement. 

To measure the upper limb strength, the load cell is fixed to the wall at one end to hold it in place, and at the other end it is attached to the participant's forearm, with a rigid band (Nite Ize, Sangle Camjam, Boulder, CO, USA) of 2.54 cm width. The handheld dynamometer is also attached to this band so that the participant can push against it. The band holding the instruments was placed six to eight centimetres proximal to the styloid process of the ulna on the anterior surface of the forearm. The mounting has, therefore, been designed so that the participant can flex their biceps against the band for a maximum voluntary isometric contraction. Standing with their backs to the wall, participants had to perform a maximal voluntary biceps contraction. The participant was seated in a chair, while measuring the lower limb strength. Participants were instructed to cross their arms at chest level and were also secured with a strap over the thigh at 8 to 10 cm proximal to the upper aspect of the patella to be stabilized, thus avoiding any extraneous movement [2]. The chair was on a raised platform so that the participants feet were not in contact with the floor. The load cell was attached to the rear legs of the chair at one end and to the participant’s leg with a band of 2.54 cm width at the other end. The handheld dynamometer was also attached to this band so that the participant would push against it. The band that held the instruments was placed six to eight centimetres proximal to the malleolus. The position of the chair was adjusted so that the cable going from the rear legs to the participant’s leg was slightly tensed when the participant's knee was bent at 90°. The position was adjusted so that the dominant leg could push against the band in maximum voluntary isometric contraction without any friction with the ground (Figure 1). The protocol consisted of four maximum voluntary isometric contractions with 60 seconds pause between each contraction for both, the upper and lower body testing. In order to standardize the movement of execution between participants, all participants received the same verbal instructions regarding how to perform the maximum isometric contraction. All of the participants were instructed in the same manner to perform the physical task (upper or lower limb contraction) when they were ready and to push as hard as possible, and to maintain this for four seconds. All of the devices were simultaneously worn by the participants and the handheld dynamometer had a feature that was activated that emitted an auditory ‘beep’ when force was detected and another ‘beep’ four seconds after the initial activation ‘beep’, signaling the end of the contraction. The participants were instructed to prepare themselves to perform a maximal effort, as soon as the handheld dynamometer detected a tension threshold of 2 kg (19.61 N), the four second interval commenced. The same protocol was repeated for the second session, which took place a minimum of 48 h after the first session. 

In this experiment the load cell output was split to be sent to two devices. From one side it was attached to an amplification tool, the HX711, to amplify the output signal making it accessible to the Arduino software to communicate the measurements to a PC via USB [17]. The Arduino software is free and open source available online for download [18]. The microcontroller-based system, combined with the Arduino software, facilitates its use and the data acquisition process. The overall system is easily implemented for those unfamiliar with this field, and the codes used to perform this experiment are also available online [17]. Furthermore, a one-point calibration of the device with weights of 24.04 kg was carried out before the first measurement of the maximum isometric strength development of the lower limb and a one-point calibration of the device with weights of 11.34 kg was carried out before the first measurement of the maximum isometric strength development of the upper limb. These selected weights were representative of the average strength of the upper and lower limbs of university students after some pilot testing was performed. Tera Term [19] (a free and open source software) was used to convert the data collected with the Arduino microcontroller into an Excel spreadsheet on a computer system in order to record the participants' results. On the other side, the load cell used was connected to a data acquisition card and the data collection was performed by a custom Matlab script. This script simultaneously allowed the data to be collected digitally. For this commercial grade signal conditioner, a two-point calibration was performed to resize the output using the same weights as for the Arduino software calibration. 

### Data Analysis

Peak values were identified for the microcontroller-based load cell amplifier and for the commercial grade signal conditioner by a custom Matlab program and were visually confirmed. For the handheld dynamometer, the peak values from each trial were instantly identified by the device and subsequently noted by hand. Then, the results were analyzed to determine the validity and reliability of the HX711 microcontroller-based load cell by comparing it to the standard laboratory setup and the handheld dynamometer. 

Once the peak values were obtained for each of the trials for each of the devices, the average of the four consecutive attempts was calculated to determine the overall strength of each participant for each device configuration, for both upper and lower limbs. All collected measurements were expressed in kilograms. To assess criterion validity between the two pairs of instruments, Pearson’s correlation analysis was used [20]. The interpretation of the correlation coefficient has been classified as follows: 0.0–0.1 being a trivial correlation, 0.1–0.3 low, 0.3–0.5 moderate, 0.5–0.7 high, 0.7–0.9 very high, and 0.9–1 a nearly perfect correlation [21]. As well as obtaining these discrete values, visual exploration of the correlations between means is illustrated by simple scatterplots with regression slopes. Visual representations of the mean difference plots are also provided to assess a level of agreement between the scores collected for each device by plotting the mean difference between two techniques against the average of the two [22], and by constructing limits of agreement within which 95% of the differences of the second method, from the first, occur [23].

The upper and lower limb strength measurements were combined when calculating the ICC for both the inter-day and intra-day reliably. To assess the inter-day reliability of the instruments, intraclass correlation coefficients (ICC) were calculated by comparing each instrument's results from the first session with those from the second session. The compared results were composed of the average lower limb strength measurement and the average upper limb strength measurement of each participant from the first and second sessions. However, the upper limb and lower limb measurements were separated in the Pearson’s correlation calculation as they represented two different conditions in the testing sessions. The ICC estimates and their 95% confident intervals used to assess inter-day reliability were calculated based on a mean-rating, absolute-agreement, two-way mixed-effects model. To assess the intra-day reliability of the instruments, intraclass correlation coefficients were also calculated however, by comparing each of the four trials among themselves for all instruments, for both sessions individually. The compared results were composed of the four lower limb MVIC trials and the four upper limb MVIC trials, separated by measurement tool and session. The ICC estimates and their 95% confident intervals used to assess intra-day reliability were calculated based on a single-rating, absolute-agreement, 2-way mixed-effects model. The ICC form selection process follows the guideline provided by Koo and Li (2016) regarding the selection of intraclass correlation coefficients for reliability research. A general threshold for the strength of the associations using the ICC method was set at excellent (0.90–1), good (0.75–0.90), average (0.50–0.75) or low (0–0.50) [24]. RStudio Version 1.3.959 was used for all statistical analysis. All statistical analyses were registered in an open platform, where the R code used is described and where more detailed information regarding the statistical analysis can be obtained [25]. 

## 3. Results

A total sample of 30 university students (27 women and 3 men) were recruited and consented to participate in the study. All 30 participants completed the study. The mean age of the participants was 21.1 years (sd = 2.3 years), mean height was 165 cm (sd = 8 cm), and mean weight was 65.1 kg (sd = 19.6 kg). 

### 3.1. Validity

Validation results for both the HX711 microcontroller-based load cell compared to the laboratory based setup, and the HX711 microcontroller-based load cell compared to the handheld dynamometer, are found in scatterplots (Figure 2, Figure 3, Figure 4 and Figure 5) with linear regression and mean difference plots (Figure 6, Figure 7, Figure 8 and Figure 9) illustrating the observed agreements between instruments. A summary description of the mean difference plots, including bias, upper limit of agreement and lower limit of agreement are provided in Table 2 and Table 3. For the correlations, the criterion validity for the instruments evaluated simultaneously was nearly perfect for every analysis that is, for the upper and lower body in the first and second sessions. Similar validity was demonstrated for the HX711 microcontroller-based load cell when compared to the standard laboratory setup and the handheld dynamometer, with Pearson's R values for the first session of 0.977 and 0.971 for the upper limb and of 0.987 and 0.971 for the lower limb. Similar Pearson’s R values were obtained for the second session with upper limb values of 0.947 and 0.969 and lower limb values of 0.992 and 0.985.

### 3.2. Reliability

Statistical analysis of inter-day reliability was conducted using the intraclass correlation coefficients and their 95% confidence intervals to compare the strength measurements obtained in two separate sessions for each tool. The results are reported in Table 4. All three tools had excellent ICC values (0.90–1) when based on a mean-rating, absolute-agreement, 2-way mixed-effects model. Statistical analysis of intra-day reliability was conducted using the intraclass correlation coefficients and their 95% confidence intervals to compare the strength measurements obtained between each of the MVIC trials for each participant in the same sessions and for each tool individually. The results obtained from the strength measurements from the first session are reported in Table 5 and the results obtained from the strength measurements from the second session are reported in Table 6. All three tools had good ICC values (0.75–0.90) for the first and second sessions when based on a single-rating, absolute-agreement, 2-way mixed-effects model.

## 4. Discussion

The objective of this study was to assess the criterion validity, inter-day reliability and intra-day reliability of a low-cost microcontroller-based load cell amplifier for measuring lower limb and upper limb maximal isometric muscular force development. A standard laboratory setup consisting of a commercial grade signal conditioner was used as a reference for laboratory-based strength measurement tools. For clinical and portable applications, a handheld dynamometer was selected for comparison purposes. The methodology used in this study tested all three tools simultaneously, that is, for the same muscle contractions. The results of this study showed nearly perfect correlation between instruments, demonstrating the validity of the HX711 microcontroller-based load cell amplifier, and good ICC scores for both the inter- and intra-day reliability assessment for all tree devices, demonstrating its reliability. 

This finding is consistent with a previous study conducted by Culter et al. (2018), who also used a load cell interfaced to a HX711 amplifier and an Arduino microcontroller, showing that the load cell compares favorably to a pressure sensing fabric and a mechanical pinch gauge in measuring pinch strength contraction. In this case, the load cell showed very high Pearson’s correlation (R = 0.83) with their commercial mechanical pinch gauge [16]. Several other studies have investigated the use of low-cost technologies, such as microcontroller-based technology. A similar system to the HX711 microcontroller-based load cell amplifier to measure muscular strength is a microcontroller-based jump mat system, with free software and open hardware, used to measure the vertical jump called Chronojump, but can also be used for measuring muscular strength. A study conducted by Pueo et al. (2018), examines the validity and reliability of the Chronojump jump mat by comparing it with proprietary systems. The study suggests that the Chronojump system provides results comparable to the proprietary systems, thereby, demonstrating its validity and reliability [26]. This is a comparable example of a low-cost system offering a potential to measure muscular strength. While, it is early in the integration of these low-cost devices into clinical and research situations, these promising early results indicate their potential. The results of this project demonstrate that a microcontroller-based device strongly correlates with the Microfet 2, a very encouraging result since the validity and reliability of the Microfet 2 has been demonstrated on several occasions. For example, a study by Mentiplay et al. (2015), looked at the reliability and the validity of handheld dynamometers for assessing lower body isometric muscular strength and power. These authors evaluated two handheld dynamometers, including the Microfet 2, against the KinCom dynamometer. Their study showed good to excellent reliability and validity with the use of handheld dynamometers [27]. A second example is a study conducted by Buckinx et al. (2017), investigating the validity of a portable device, the Microfet 2 handheld dynamometer, for measuring maximal isometric voluntary contraction in an elderly population. The results of this study showed for all muscle groups, with the exception of the ankle, high relative and moderate absolute reliability [28], thus, demonstrating the reliability of the Microfet 2 as a tool to asses isometric strength. These are encouraging as these studies are in agreement with the conclusions in the present study, and therefore supporting the validity and reliability not only for the Microfet 2 handheld dynamometer, but also for the HX711 microcontroller-based load cell amplifier, due to highly similar force measurements obtained with both devices.

Microcontroller-based load cell amplifier devices used to measure upper and lower limb muscle strength have many advantages. In general, they are low-cost and highly customizable, while offering maximum strength values similar to those of standard devices. Regarding custom applications, the specific handheld dynamometer used in this study displays force in increments of 0.1 kg (0.98 N), while the microcontroller-based load cell amplifier can be programmed to a sensitivity of force variations of less than 0.1kg (0.98 N) as in the study by Culter et al. (2018). Secondly, in comparison with the standard laboratory setup, the microcontroller-based load cell amplifier is a small portable device that is compact. It can be easily transported from one location to another, making it easier to assess muscle strength in people with limited mobility and opening the possibility for laboratory grade field strength testing. Although, many benefits have been identified, this study has some limitations that may reduce the generalizability of the results. This study recruited young, healthy university students as participants. Due to the largely homogeneous sample, the results may not be generalizable to the general population. A second limitation of this study concerns the reliability of the instruments. The results between trials and sessions may vary due to psychological factors such as the habituation factor. In the first trial of the first session, participants may not know what to anticipate despite the given explanations, and therefore, become increasingly comfortable with the instruments as the trials progress. Furthermore, even with the evaluation protocol being identical for both testing sessions, strength values in the separated sessions might differ due to reasons out of the evaluators’ control, such as fatigue due to other daily activities of the participant. Moreover, with the protocol being identical for each participant, internal motivation is also a potential limitation as some participants may have seen their score on the handheld dynamometer and may have used it as a motivation to aim for an increase in future trials scores. However, these values were not explicitly shown to participants. Despite these limitations, our results demonstrate strong reliability between sessions and strong intra-day reliability.

## 5. Conclusions

The results of this study could have significant implications for the way in which the evaluation of muscle strength measurement is performed in the future. The quantification of muscle strength is an essential aspect of physical capacity assessment, and the use of the HX711 microcontroller-based load cell amplifier, as described in this study, could be a useful substitute to current methods. The HX711 microcontroller-based load cell amplifier shows nearly perfect correlation with the standard laboratory setup and the handheld dynamometer, which demonstrates its validity, and shows a good association when calculating the inter-day and intra-day reliability score with the use of the ICC, which demonstrates its reliability. There are numerous possibilities in terms of practical implications for the microcontroller-based load cell amplifier and further studies should evaluate its use in laboratory settings and clinical settings. Having characteristics, including commercial-like grade quality at a low cost, as well as being small and portable, this device has the potential to be used in a wide range of settings.

## Figures and Tables

**Figure 1 sensors-20-04999-f001:**
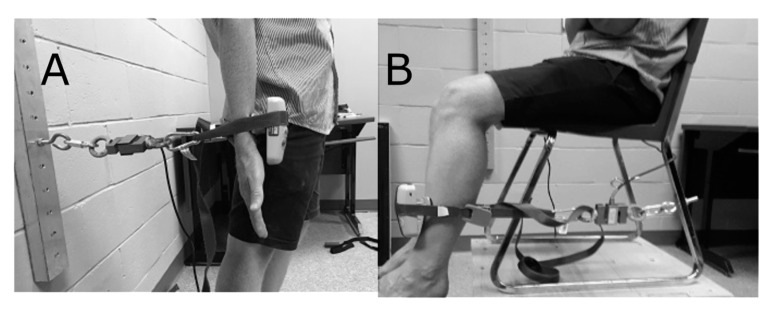
Experimental setup for simultaneously measuring all methods of force measurements for the upper; (**a**) and lower; (**b**) limbs.

**Figure 2 sensors-20-04999-f002:**
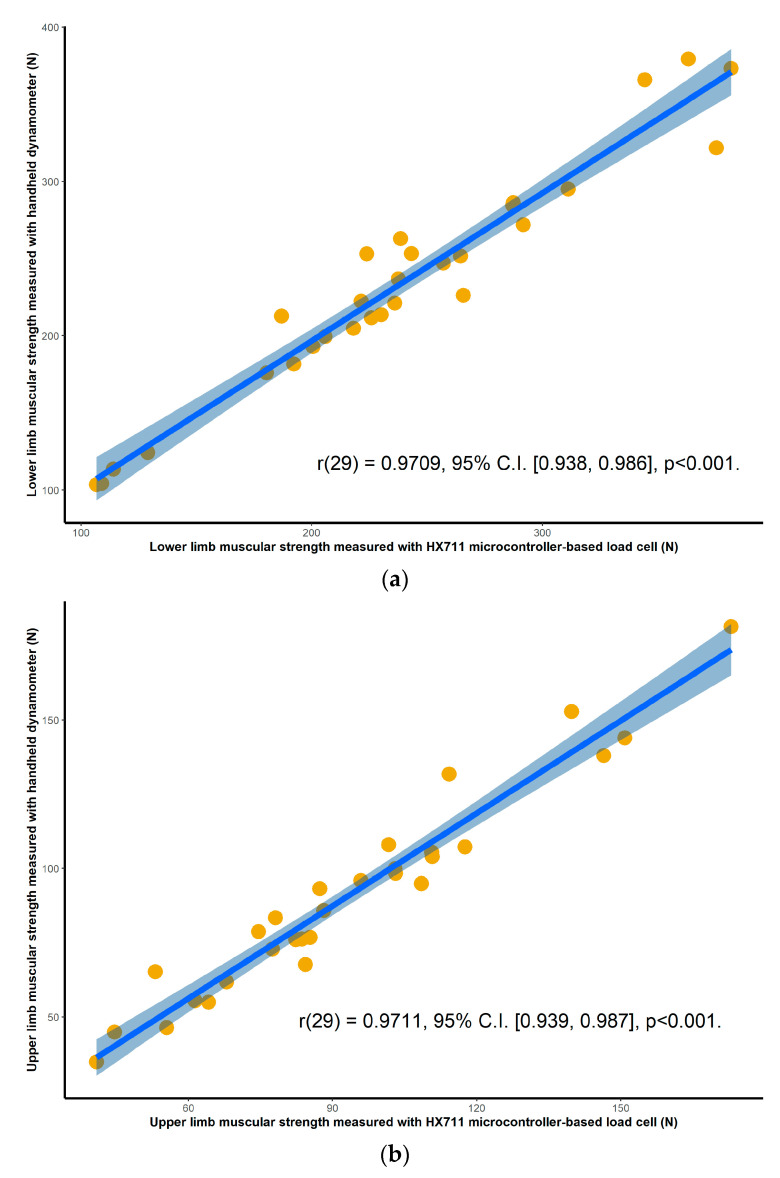
Session one correlations between the handheld dynamometer and the HX711 microcontroller-based load cell for: (**a**) maximal isometric leg extension; (**b**) maximal isometric bicep flexion. The blue line and the shaded area are the linear regression slope and its 95% confidence region.

**Figure 3 sensors-20-04999-f003:**
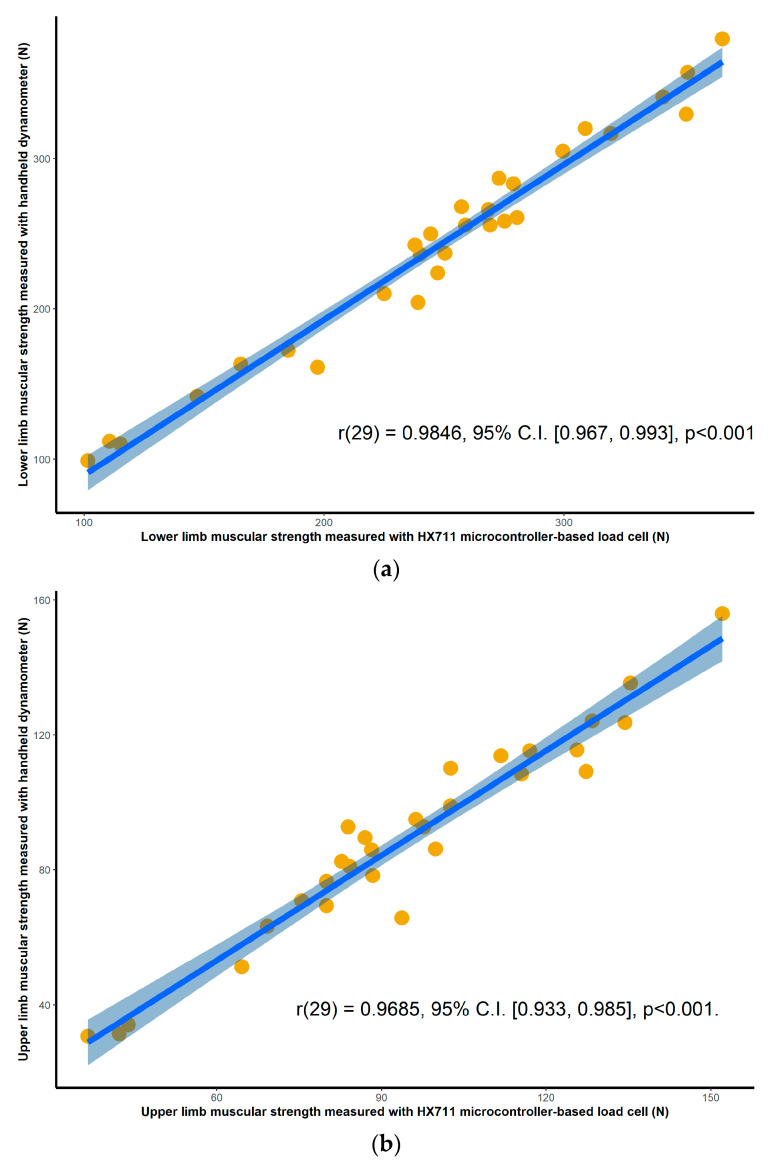
Session two correlations between the handheld dynamometer and the HX711 microcontroller-based load cell for: (**a**) maximal isometric leg extension; (**b**) maximal isometric bicep flexion. The blue line and the shaded area are the linear regression slope and its 95% confidence region.

**Figure 4 sensors-20-04999-f004:**
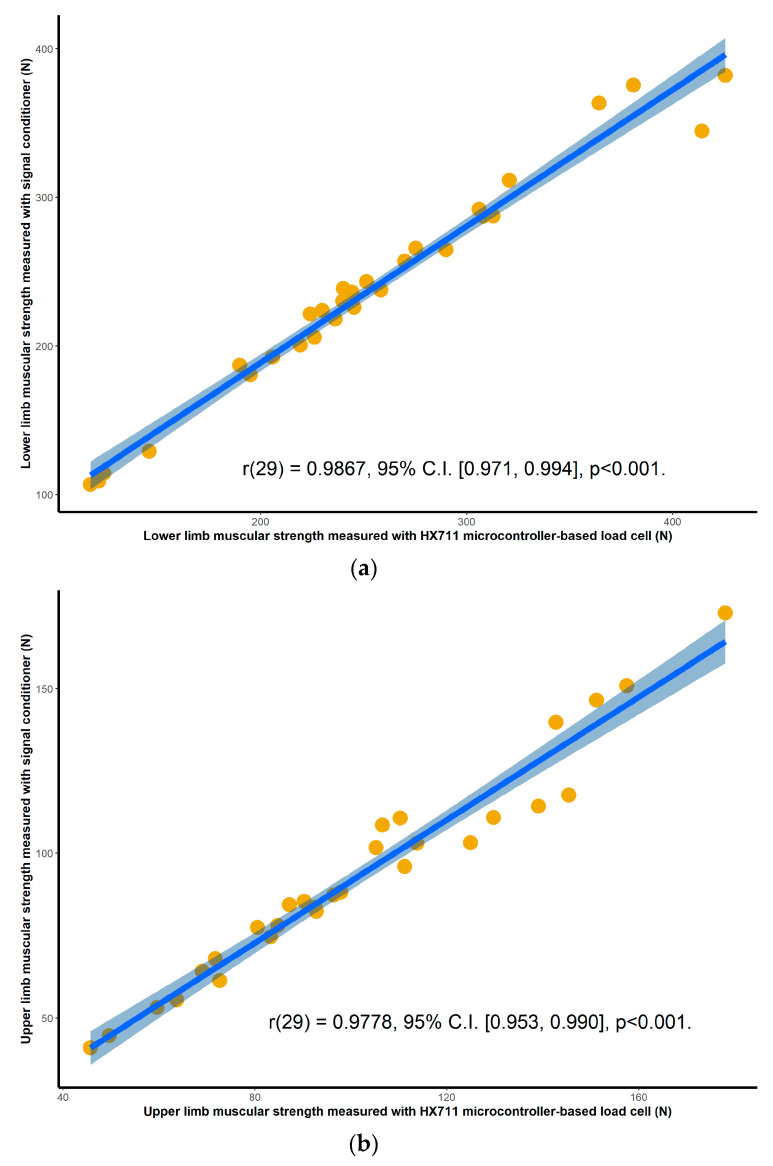
Session one correlations between the commercial signal conditioner and the HX711 microcontroller-based load cell for: (**a**) maximal isometric leg extension; (**b**) maximal isometric bicep flexion. The blue line and the shaded area are the linear regression slope and its 95% confidence region.

**Figure 5 sensors-20-04999-f005:**
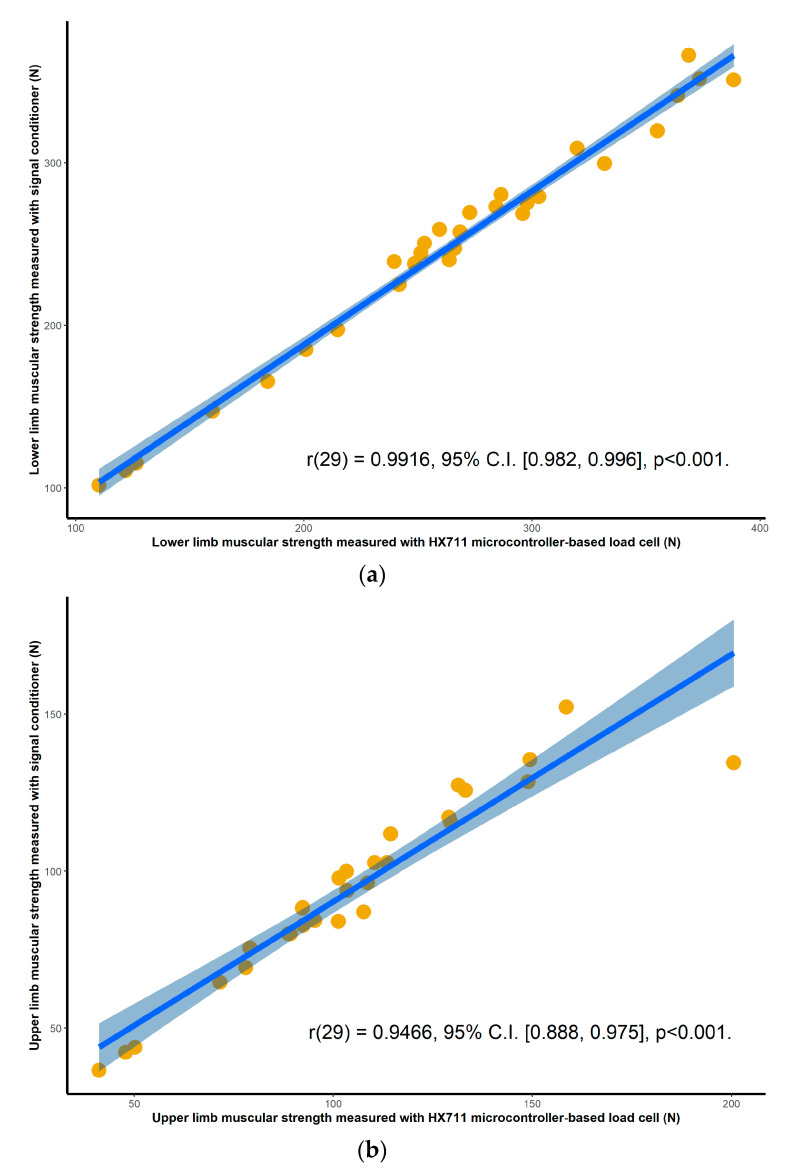
Session two correlations between the commercial signal conditioner and the HX711 microcontroller-based load cell for: (**a**) maximal isometric leg extension; (**b)** maximal isometric bicep flexion. The blue line and the shaded area are the linear regression slope and its 95% confidence region.

**Figure 6 sensors-20-04999-f006:**
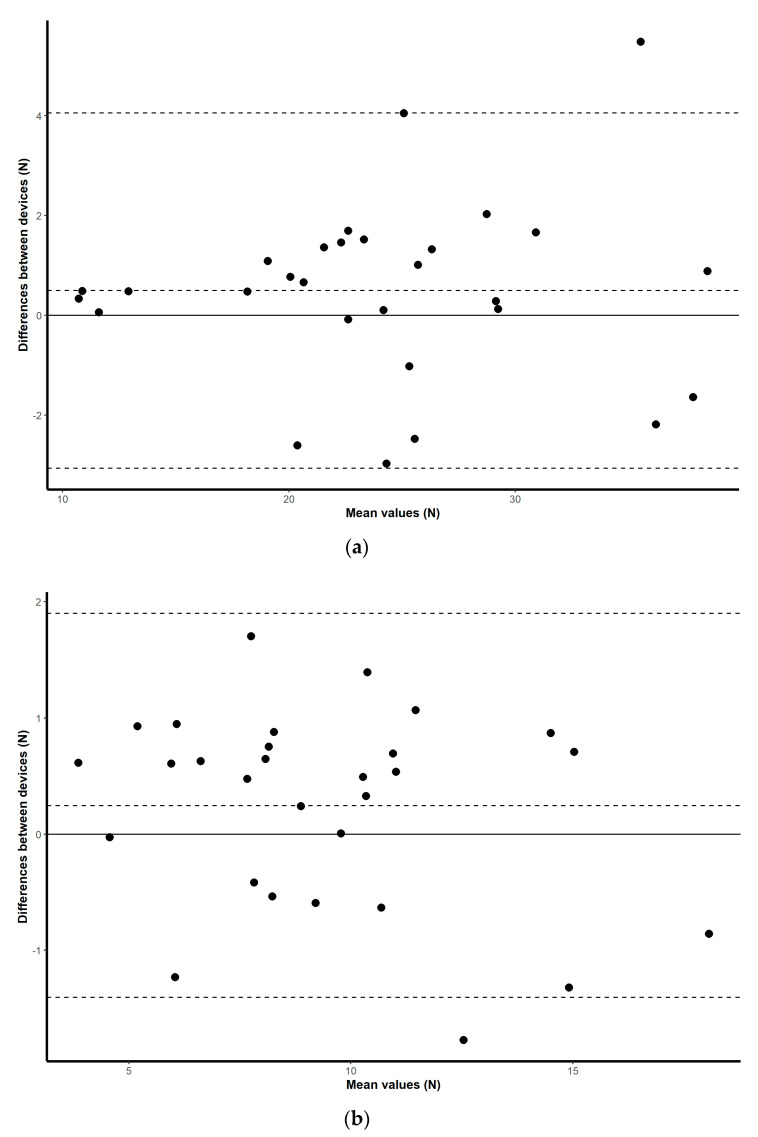
Session one mean difference plot comparison of the results from the HX711 microcontroller-based load cell and the handheld dynamometer for: (**a**) maximal isometric leg extension; (**b**) maximal isometric bicep flexion. The middle-dashed line represents the mean difference and the upper and lower dashed lines represent the 95% limits of agreement (mean ± 1.96 * SD).

**Figure 7 sensors-20-04999-f007:**
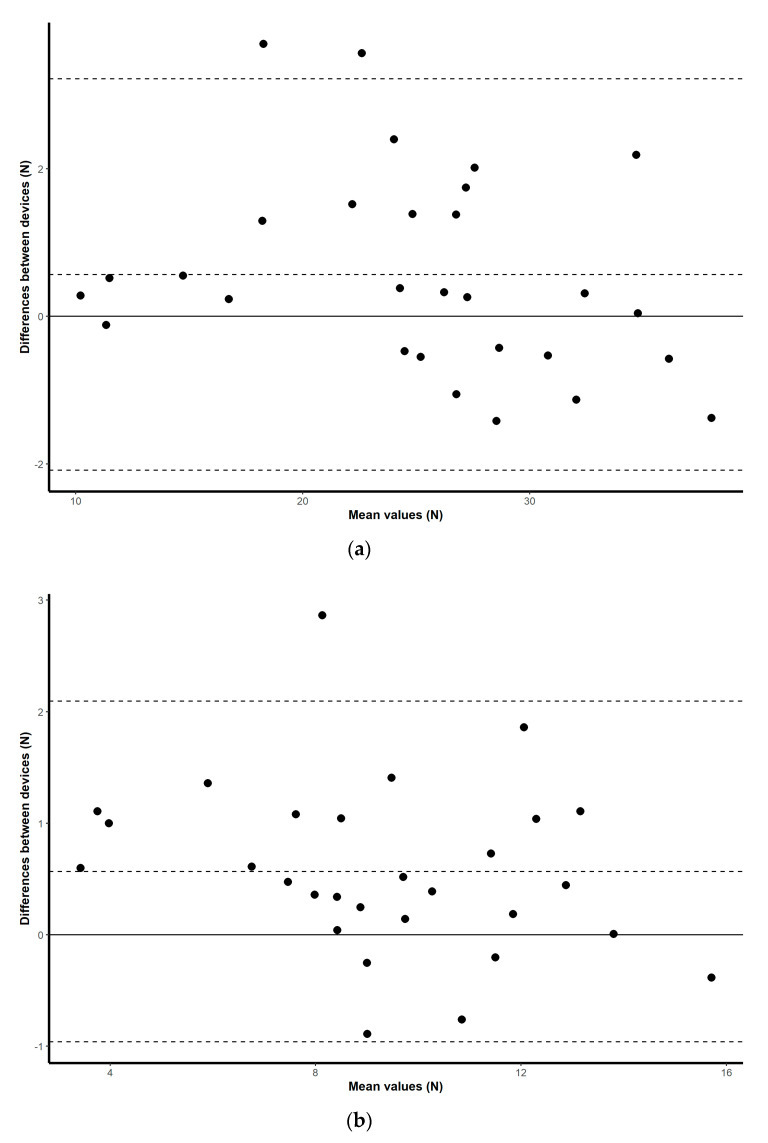
Session two mean difference plot comparison of the results from the HX711 microcontroller-based load cell and the handheld dynamometer for: (**a**) maximal isometric leg extension; (**b**) maximal isometric bicep flexion. The middle-dashed line represents the mean difference and the upper and lower dashed lines represent the 95% limits of agreement (mean ± 1.96 * SD).

**Figure 8 sensors-20-04999-f008:**
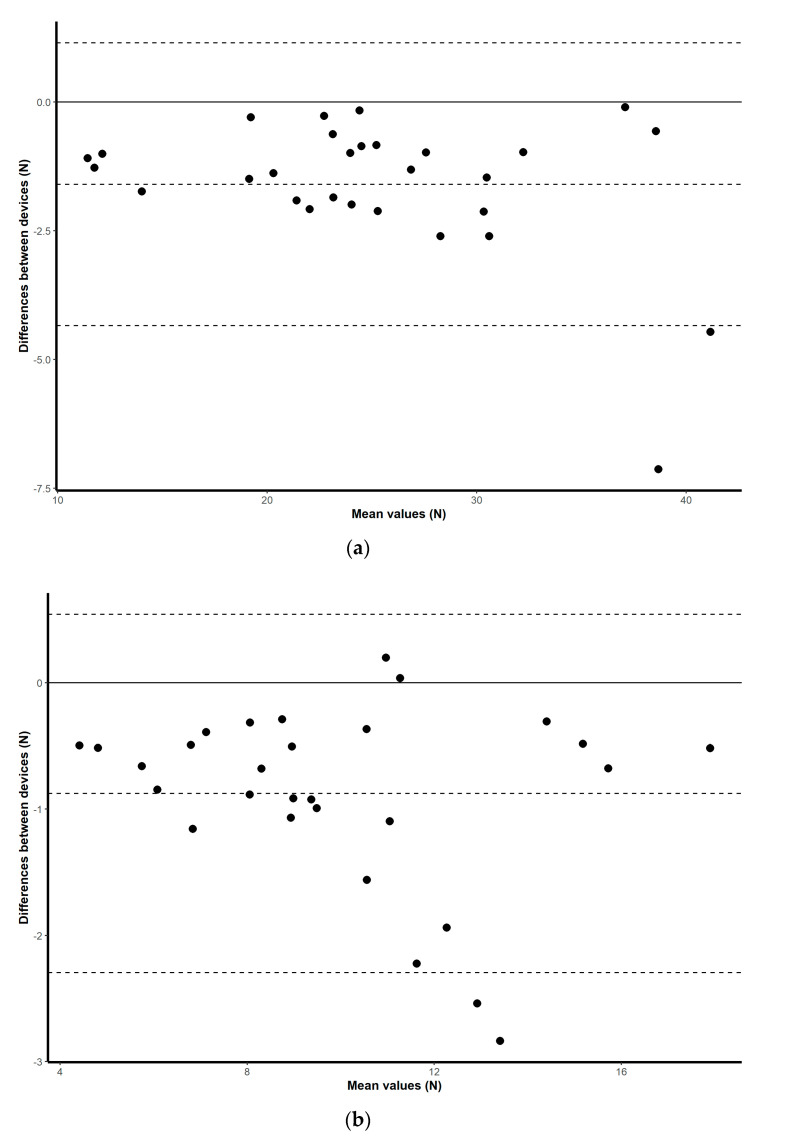
Session one mean difference plot comparison of the results from the HX711 microcontroller-based load cell and the signal conditioner for: (**a**) maximal isometric leg extension; (**b**) maximal isometric bicep flexion. The middle-dashed line represents the mean difference and the upper and lower dashed lines represent the 95% limits of agreement (mean ± 1.96 * SD).

**Figure 9 sensors-20-04999-f009:**
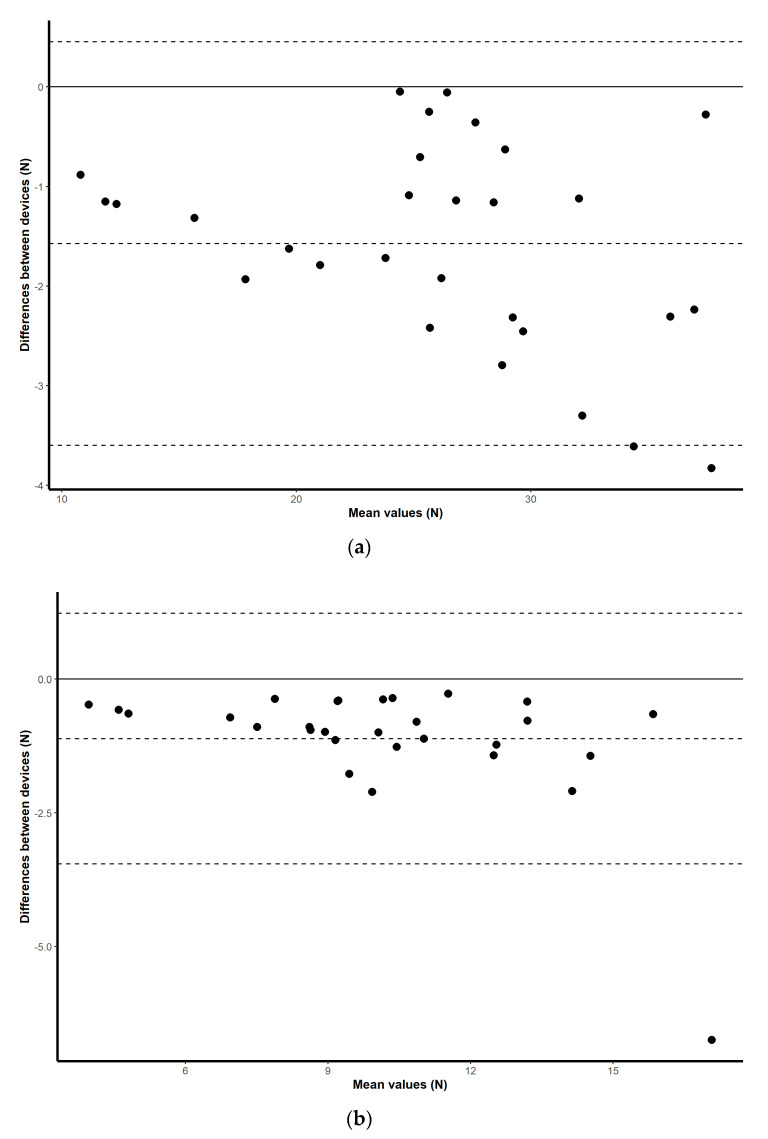
Session two mean difference plot comparison of the results from the HX711 microcontroller-based load cell and the signal conditioner for: (**a**) maximal isometric leg extension; (**b**) maximal isometric bicep flexion. The middle-dashed line represents the mean difference and the upper and lower dashed lines represent the 95% limits of agreement (mean ± 1.96 * SD).

**Table 1 sensors-20-04999-t001:** A relative comparison between the three different instruments (Lafayette handheld dynamometer, the commercial signal conditioner and the HX711 microcontroller-based load cell amplifier) used for measuring strength.

Characteristics	HX711 (+Arduino UNO)	Lafayette Handheld Dynamometer	Signal Conditioner
Relative cost	+	++	+++
Weight	+	++	+++
Size (cm)	+	++	+++
Power supply	Battery and wall mount adapter	Battery	Wall mount adapter
Range	Customizable	Non-customizable	Customizable
Calibration points	Customizable	Non-customizable	Customizable
Resolution	Customizable	Non-customizable	Customizable

**Table 2 sensors-20-04999-t002:** Session one summary of the mean difference comparison results between two pairs of instruments for maximal isometric leg extension (lower limb) and maximal isometric bicep flexion (upper limb), including the mean difference, and the upper and lower 95% limits of agreement (mean ± 1.96 * SD).

Condition		Lower Limb			Upper Limb		
		Estimate	Lower CI	Upper CI	Estimate	Lower CI	Upper CI
HX711 vs. signal conditioner	Bias Lower 95% limit of agreement Upper 95% limit of agreement	−1.5976 −4.3401 1.1450	−2.1298 −5.2604 0.2247	−1.0653 −3.4199 2.0652	−0.8774 −2.2936 0.5388	−1.1523 −2.7688 0.0636	−0.6026 −1.8184 1.0140
HX711 vs. HHD	Bias Lower 95% limit of agreement Upper 95% limit of agreement	0.4938 −3.0640 4.0516	−0.1966 −4.2577 2.8578	1.1843 −1.8702 5.2454	0.2464 −1.4061 1.8989	−0.0743 −1.9605 1.3444	0.5671 −0.8516 2.4533

**Table 3 sensors-20-04999-t003:** Session two summary of the mean difference comparison results between two pairs of instruments for maximal isometric leg extension (lower limb) and maximal isometric bicep flexion (upper limb), including the mean difference, and the upper and lower 95% limits of agreement (mean ± 1.96 * SD).

Condition		Lower Limb			Upper Limb		
		Estimate	Lower CI	Upper CI	Estimate	Lower CI	Upper CI
HX711 vs. signal conditioner	Bias Lower 95% limit of agreement Upper 95% limit of agreement	−1.5732 −3.5972 0.4509	−1.9660 −4.2764 −0.2282	−1.1803 −2.911 1.1301	−1.1120 −3.4517 1.2277	−1.5661 −4.2368 0.4426	−0.6579 −2.6666 2.0128
HX711 vs. HHD	Bias Lower 95% limit of agreement Upper 95% limit of agreement	0.5673 −2.0849 3.2194	0.0526 −2.9748 2.3295	1.0820 −1.1950 4.1093	0.5679 −0.9593 2.0950	0.2715 −1.4717 1.5826	0.8643 −0.4468 2.6075

**Table 4 sensors-20-04999-t004:** Results of ICC calculation comparing each instrument’s from the first session with those from the second session, and their 95% confident intervals, using a mean-rating, absolute-agreement, 2-way mixed-effects model.

Tool	Value	Conf.level	Lbound	Ubound	F-Value	DF1	DF2	*p*-Value
HX711	0.9582	0.95	0.9297	0.9752	24.0541	57	57.9480	*p* < 0.001.
HHD	0.9583	0.95	0.9298	0.9753	23.8153	57	57.7598	*p* < 0.001.
Signal conditioner	0.9528	0.95	0.9204	0.9720	21.3221	57	57.8473	*p* < 0.001.

**Table 5 sensors-20-04999-t005:** Session one results of ICC calculation comparing each participant’s MVIC trials among themselves for each instrument and their 95% confident intervals, using a single-rating, absolute-agreement, 2-way mixed-effects model.

Tool	Value	Conf.level	Lbound	Ubound	F-Value	DF1	DF2	*p*-Value
HX711	0.9269	0.95	0.8909	0.9533	56.3835	56	124.2922	*p* < 0.001.
HHD	0.9169	0.95	0.8769	0.9468	48.0088	54	138.3623	*p* < 0.001.
Signal conditioner	0.9208	0.95	0.8813	0.9495	52.4854	56	114.3274	*p* < 0.001.

**Table 6 sensors-20-04999-t006:** Session two results of ICC calculation comparing each participant’s MVIC trials among themselves for each instrument and their 95% confident intervals, using a single rating, absolute-agreement, 2-wat mixed-effects model.

Tool	Value	Conf.level	lbound	ubound	F-Value	DF1	DF2	*p*-Value
HX711	0.9135	0.95	0.8726	0.9441	46.5856	57	134.9212	*p* < 0.001.
HHD	0.9175	0.95	0.8781	0.9468	49.0946	57	132.8442	*p* < 0.001.
Signal conditioner	0.9184	0.95	0.8797	0.9473	49.4105	57	137.3947	*p* < 0.001.

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
