# Peer review of "Assessing the Validity and Reliability of A Low-Cost Microcontroller-Based Load Cell Amplifier for Measuring Lower Limb and Upper Limb Muscular Force"

_sensors, 2020, doi:10.3390/s20174999_

Round 1

Reviewer 1 Report

This is a simple, well written study. I recommend publication, however would like to see additional information about the comparison between the HX711 and the lab amplifier by way of variables such as mean and maximum error between the devices.

Author Response

REVIEWER 1:

This is a simple, well written study. I recommend publication, however, would like to see additional information about the comparison between the HX711 and the lab amplifier by way of variables such as mean and maximum error between the devices.

We thank the reviewer for the kind general comment on the manuscript. In response to the request for additional information such as the mean and the maximum error between the devices, tables summarizing the mean difference plots have been added. These tables include the mean difference (bias) and it’s lower and upper 95% confidence interval, the lower 95% limit of agreement (e.g., -1.96 sd) and it’s lower and upper 95% confidence interval, and the upper 95% limit of agreement (e.g., +1.96 sd) and it’s 95% upper and lower confidence interval. These tables are identified as table 1 and table 2 in the manuscript. We feel that the inclusion of this information is helpful for comparison purposes and potentially adjustment for mean and maximum error between devices for future studies. We thank the reviewer for this comment as it improves the manuscript.

Reviewer 2 Report

Good study presenting data about a low-cost microcontroller to assess muscular strength. Some concerns and clarifications below:

ABSTRACT

Clear and concise.

INTRODUCTION

Muscle strength is more than the authors describe in the first paragraph. A lot of strength forms are omitted. Please, describe muscle strength in a broader way.

A part of the test performed, the rest interval between two consecutive repetitions, the number of repetitions performed, and the body position. There are other several factors influencing the reliability of MVIC. For instance, the explosiveness of the muscular contraction.

I suggest to include more microcontrolled sensors to assess muscular strength such as Chronojump, Musclelab and other systems of different cost ranges. There are a lot in the market.

METHODS

Please, include a picture of the microcontrolled system tested and another picture of the testing set up.

Which method is the golden standard? Please, clarify.

I don’t understand how the Microfet 2 test and the load cell were carried out simultaneously. How did you used the Microfet 2 and displayed a full resistance? Please clarify.

Why did you choose two days to assess the reliability? I’m interested in the intra-session reliability. The strength values between the two sessions might differ because of other reasons. For instance, fatigue, other activities, intakes, …

RESULTS

The characteristics of the sample should be placed in the methods section.

Page 6 line 215. Please correction is necessary.

DISCUSSION

The first paragraph is a too long summary of the results that the reader has just read. In my opinion, is not necessary.

There is a very similar system called Chronojump that should to be mentioned in the discussion, instead of EMG low-cost system. I don’t understand the connection.

In my opinion, the discussion state characteristics of the instrument that has to be mentioned in the methods section. The statement that is low-cost and accessible is too redundant. However, no mentions about the accessibility of the software, its usability and the data reports/plots.

In respect to reliability data, the authors present the motivation as a limitation. This should to be controlled and equally provided to all participants.

There are a lot of studies assessing the validity and reliability of Microfet 2 and gadgets like this. I think this should be mentioned in the discussion section in order to compare your results.

REFREENCES

Inconsistent style in referencing. Please, follow the rules.

Author Response

REVIEWER 2:

Good study presenting data about a low-cost microcontroller to assess muscular strength. Some concerns and clarifications below:

Thank you for the kind general comment on the manuscript. We provide a response to each of the concerns and clarifications below.

ABSTRACT

Clear and concise.

We made no modifications to the abstract.

INTRODUCTION

Muscle strength is more than the authors describe in the first paragraph. A lot of strength forms are omitted. Please, describe muscle strength in a broader way.

We thank the reviewer for this comment. In response, we have added the following information to the introduction of our manuscript:

“Muscle strength has been defined as the maximum force (in N) or torque (in Nm) developed during maximal voluntary contraction under a given set of conditions. Thus, the term “muscular strength” typically refers to a measure describing an individual’s ability to exert maximal muscular force, either statically or dynamically. In all circumstances, an activated muscle develops force. Depending on the interaction between the force developed by the muscle and the load on the muscle, the muscle will either shorten, remain at a fixed length, or be lengthened. Tension development without changing the length of the muscle is an isometric muscle action; the muscle acts against an immovable resistance at a specific joint angle. Tension development with changes in the length of the muscle, allowing for a complete range of motion, is an isotonic muscle action. An isotonic-concentric action involves the development of tension during shortening of the muscle, whereas an isotonic-eccentric action occurs when there is development of tension during lengthening of the muscle.”

As you can see, we have broadened the description of muscle strength to more widely represent the potential applications of measuring muscle strength.

A part of the test performed, the rest interval between two consecutive repetitions, the number of repetitions performed, and the body position. There are other several factors influencing the reliability of MVIC. For instance, the explosiveness of the muscular contraction.

We thank the reviewer for this comment as this is an important consideration. The reviewer is correct in identifying that standardizing the experimental setup and task procedures is important. In order to standardize the movement execution between participants, we performed our measures in the following manner. All of the participants were instructed in the same manner to perform the physical task (upper or lower limb contraction) when they were ready and to push as hard as possible, and to maintain this for four seconds. All of the devices were simultaneously worn by the participants and the handheld dynamometer had a feature that was activated that emitted a ‘beep’ when force was detected and another ‘beep’ four seconds after the initial activation ‘beep’. The participants were instructed to prepare themselves to perform a maximal effort, as soon as the handheld dynamometer detected tension a threshold of 2kg, the four second interval commenced. In order to clearly explain this in our article, we made inserted the following information into the manuscript:

Lines 155-164 in the manuscript contain this description.

I suggest to include more microcontrolled sensors to assess muscular strength such as Chronojump, Musclelab and other systems of different cost ranges. There are a lot in the market.

We thank the reviewer for this comment. Certainly, there is a proliferation of such devices on the market, with likely more to follow soon. This is in fact a reason why we are interested in this technology, there is a tremendous potential to change current kinetic and kinematic measurement practices with such low-cost sensors. Also, we thank the reviewer for providing the names of two additional tools that could be used to compare against the HX711. We have future plans to continue this work, however, we do not want to delay the publication of this information as we feel that it could be beneficial to other users who want to explore the HX711 in their custom applications for measuring strength. In addition, the funding for this project is completed and there are no funds available at this time to pursue the research. We are actively searching for additional funding to continue this line of research.

METHODS

Please, include a picture of the microcontrolled system tested and another picture of the testing set up.

We thank the reviewer for this comment. We agree that including these images in the manuscript will help the reader understand what was performed during the experimental setup and data collection sessions. We have included a picture of a representative data collection setup for the upper and lower limbs in the manuscript. This is figure 1 and can be found following the first paragraph in the methods section. Concerning a picture of the microcontrolled system, we opted to not include a picture of this setup as these images are widely available on the internet and our setup did not modify the typical usage of this apparatus. Furthermore, with the additional image of the experimental setup now have nine figures and three tables in this article.

Which method is the golden standard? Please, clarify.

We thank the reviewer for this question, as this is an important point to emphasize. In response, we have added the following information to the introduction of our manuscript:

“Fortunately, there are tools that provide more objective and accurate results. Referred to as the “golden standard”, are the isokinetic dynamometers (ID) such as the Cybex II isokinetic dynamometer. However, their clinic use is limited due to their high cost, nonportability, and their complicated setup and operation. Similarly, the load cell coupled with a signal conditioner is a laboratory setup commonly used to accurately measure isometric muscle strength. To record these values an analog to digital setup and a computer are necessary, thus also making this method expensive and not portable. Fortunately, today's technological advances open the door to many portable, fast and easy to perform methods for quantifying muscle strength. A commonly used compromise between MMT and ID devices is a portable device known as the handheld dynamometer, a clinical instrument that quantifies isometric strength.”

We have also added the following sentence to the method of our manuscript:

“In this experiment, the load cell coupled with the commercial signal conditioner is referred to as the laboratory setup and holds the standard measurements of all three devices.”

As you can see, more information on the “golden standard” and description have been added to clarify which method is considered the “golden standard”, but that is has been adjusted to suit our use of a reference measurement device that is specific to our research project and the information contained in the manuscript.

I don’t understand how the Microfet 2 test and the load cell were carried out simultaneously. How did you used the Microfet 2 and displayed a full resistance? Please clarify.

We thank the reviewer for this comment. Also, we understand your confusion as in this case the adage “a picture is worth a thousand words” aptly describes the situation. To improve this information, we have included pictures (figure 1) of the experimental setup. As you can see, the experimental setup permitted including the Microfet 2 to be securely attached to the participant simultaneously as the load cell, with the load cell output split between the HX711 amplifier and the laboratory based signal conditioner.

Why did you choose two days to assess the reliability? I’m interested in the intra-session reliability. The strength values between the two sessions might differ because of other reasons. For instance, fatigue, other activities, intakes, …

We thank the reviewer for the comment as this is an important consideration. The reviewer is correct in identifying that the strength values between two sessions might differ because of other reasons. We have therefore modified the limits of this study accordingly. We also performed more statistical analyses on the reliability of the instruments by adding an intra-session reliability assessment. The ICC model used for this added analysis is a single-rating, absolute-agreement, 2-way mixed-effects model. Furthermore, while performing this analysis we noticed an error that was performed on the inter-session reliability analysis as we have noticed a mis selection of ICC model. The ICC model used to observe inter-session reliability has been changed to a mean-rating, absolute-agreement, 2-way mixed-effects model. Previously, we had selected “single” measurement for the model when in fact it was the mean values of 4 single trials that was used. This has now been corrected throughout the text including in the results table 3, and the resume. In summary, the results of the inter-session reliability analysis have been updated and the results of the intra-session reliability analysis have been added to the results section of our manuscript in table 4 and table 5. We thank the reviewer for this comment as it improves the manuscript as it provides additional information on the reliability of the HX711 compared to the HHD and the laboratory-based signal conditioner.

RESULTS

The characteristics of the sample should be placed in the methods section.

Page 6 line 215. Please correction is necessary.

We thank the reviewer for this comment. We acknowledge that the reviewer indicates that the characteristics of the sample be included in the methods section. However, we respectfully disagree with the reviewer on this point. We do feel that certain characteristics of the sample, such as recruitment methods and exclusion criteria, should be presented in the methods section. However, we have chosen to follow an approach presented in the literature (Kotz & Cals, 2013) that indicates this information be placed in the results. With this approach, it is recommended that we should refrain from putting results in the methods section, such as the number of subjects recruited. While we thank the reviewer for bringing this to our attention, we have decided not to make changes to this section of the manuscript.

Kotz, D., & Cals, J. W. L. (2013). Effective writing and publishing scientific papers, part V: Results. Journal of Clinical Epidemiology, 66(9), 945. https://doi.org/10.1016/j.jclinepi.2013.04.003

DISCUSSION

The first paragraph is a too long summary of the results that the reader has just read. In my opinion, is not necessary.

We thank the reviewer for this comment. In response, we have removed some redundant information. The first paragraph of the introduction has been reduced significantly in size, allowing for a lighter reading of the text. We thank the reviewer as it now reads more fluently, thus improving our manuscript.

There is a very similar system called Chronojump that should to be mentioned in the discussion, instead of EMG low-cost system. I don’t understand the connection.

We thank the reviewer for this comment. We also thank the reviewer for providing pertinent information regarding a similar system. We have looked over the existing research on the Chronojump system and have decided to remove the comparison of our HX711 system to the low-cost EMG system and have replaced it with a comparison to the Chronojump system. We feel it is a better fitting system to our research. In response, we have added the following information to our manuscript:

“A similar system to the HX711 microcontroller-based load cell amplifier to measure muscular strength is a microcontroller-based jump mat system, with free software and open hardware, used to measure the vertical jump called Chronojump, but can also be used for measuring muscular strength. A study conducted by Pueo et al (2018), examines the validity and reliability of the Chronojump jump mat by comparing it with proprietary systems. The study suggests that the Chronojump system provides results comparable to the proprietary systems, thus demonstrating its validity and reliability. This is a comparable example of a low-cost system offering a potential to measure muscular strength.”

We thank the reviewer for this comment as it improves the manuscript.

In my opinion, the discussion state characteristics of the instrument that has to be mentioned in the methods section. The statement that is low-cost and accessible is too redundant. However, no mentions about the accessibility of the software, its usability and the data reports/plots.

We thank the reviewer for this comment. We have added the following information to the methods section of our manuscript:

“The Arduino software is free and open source available for download online [18]. The microcontroller-based system, combined with the Arduino software, facilitates its use and the data acquisition process. The overall system is easily implemented for those unfamiliar with this field, and the codes used to perform this experiment are also available online [17].”

“All statistical analyses were registered in an open platform, where the R code used is described and where more detailed information regarding the statistical analysis can be obtained [24].

We thank the reviewer for this comment as it improves the manuscript

In respect to reliability data, the authors present the motivation as a limitation. This should to be controlled and equally provided to all participants.

We thank the reviewer for this comment. The reviewer is right when saying that motivation should be and equally provided to all participants. Since we did provide equal motivation to all participants, we decided to bring changes to the limitation segment of our manuscript to include a more accurate description of this potential limitation. It is possible that internal motivation was potentially different between participants, in the event that they saw their scores on the Microfet from the first session and aimed for an increased score on the second visit. However, the values were not explicitly shown to the participants and therefore few participants would have seen these values. That said, we are not able to accurately state how many participants this might apply to; therefore, we continue to include this as a possible limitation. We have reworded the text on motivation in the limitations section to accurately reflect this situation. In addition, we added fatigue resulting from daily activities of the participants, therefore a possible limitation that might affect the results of our data.  

The following sentences were added to the discussion of our manuscript:

“Also, although with the protocol being identical for each participant, internal motivation is also a potential limitation as some participants may have seen their score on the handheld dynamometer and may have used it as a motivation to aim for an increase in future trials scores. However, these values were not explicitly shown to participants.”

“Furthermore, even with the evaluation protocol being identical for both testing sessions, strength values obtained in the separated sessions might differ due to fatigue related to the participants other daily activities.”  

There are a lot of studies assessing the validity and reliability of Microfet 2 and gadgets like this. I think this should be mentioned in the discussion section in order to compare your results.

We thank the reviewer for this comment. In response, we have added the following information to the discussion of our manuscript:

“The results of this project demonstrate that a microcontroller-based device strongly correlates with the Microfet 2, a very encouraging result since the validity and reliability of the Microfet 2 has been demonstrated on several occasions. For example, a study by Mentiplay and al. (2015), looked at the reliability and the validity of handheld dynamometers for assessing lower body isometric muscular strength and power. These authors evaluated two handheld dynamometers, including the Microfet 2, against the KinCom dynamometer. Their study showed good to excellent reliability and validity with the use of handheld dynamometers. A second example is a study conducted by Buckinx & al. (2017), investigating the validity of a portable device, the Microfet 2 handheld dynamometer, for measuring maximal isometric voluntary contraction in an elderly population. The results of this study showed for all muscle groups, with the exception of the ankle, high relative and moderate absolute reliability, thus, demonstrating the reliability of the Microfet 2 as a tool to asses isometric strength. These are encouraging as these studies are in agreement with the conclusions in the present study conclusions, and therefore supporting the validity and reliability not only for the Microfet 2 handheld dynamometer, but also for the HX711 microcontroller-based load cell amplifier, due to highly similar force measurements obtained with both devices.”

We thank the reviewer for this comment as it improves the manuscript.

REFREENCES

Inconsistent style in referencing. Please, follow the rules.

We thank the reviewer for their attention to detail. We have modified the references so that they are standardized and follow the journal reference format for Sensors.

Round 2

Reviewer 2 Report

Thanks for considering the review. The manuscript has improved. 

Author Response

Thank you for providing the comments that helped us improve the manuscript.